# Motor Coordination and Global Development in Subjects with Down Syndrome: The Influence of Physical Activity

**DOI:** 10.3390/jcm11175031

**Published:** 2022-08-27

**Authors:** Marianna Alesi, Valerio Giustino, Ambra Gentile, Manuel Gómez-López, Giuseppe Battaglia

**Affiliations:** 1Department of Psychology, Educational Science and Human Movement, University of Palermo, 90128 Palermo, Italy; 2Sport and Exercise Sciences Research Unit, Department of Psychology, Educational Science and Human Movement, University of Palermo, 90144 Palermo, Italy; 3Department of Physical Activity and Sport, Faculty of Sports Sciences, University of Murcia, Santiago de la Ribera, 30720 Murcia, Spain; 4Regional Sports School of CONI Sicilia, 90141 Palermo, Italy

**Keywords:** motor skills, motor coordination, motor development, global development, physical activity, Down Syndrome

## Abstract

Background: Many research studies have investigated motor impairments and delayed development in children with Down Syndrome (DS). However, very few studies detected these features in adults with DS. Hence, this study aimed to investigate the relationship between motor coordination and global development in subjects with DS, including adults. Furthermore, the second aim was to detect any differences in motor coordination and global development as a function of the practice of physical activity (PA) in this population. Methods: Twenty-five participants with DS (10 f, 15 m), with a chronological mean age of 27.24 years and development mean age of cognitive area of 4.93 years, were enrolled and divided into a physically active group (PA-G; *n* = 15) and a physically inactive group (PI-G; *n* = 10). All participants performed the Movement Assessment Battery for Children (M-ABC) to assess fine and gross motor skills, while the Developmental Profile 3 (DP-3) checklist was administered to the parents in order to screen strengths and weaknesses of five developmental areas of their relatives with DS. Results: Our results showed positive correlations between the following variables: global motor coordination and global development, global motor coordination and adaptive behavior development area, aiming and catching skills and global development, aiming and catching skills and adaptive behavior development area. As for the practice of PA, PA-G showed higher scores than PI-G in all the tasks of both the M-ABC and the DP-3, though significant differences were found only for the global motor coordination, for the aiming and catching skills, as well as for the physical development area. Conclusions: The findings of this study reinforce the need to implement and encourage the practice of PA in order to promote well-being and social inclusion in subjects with DS.

## 1. Introduction

Research has amply demonstrated a close relationship between Intellectual Quotient (IQ) and motor proficiency, showing that people with mild or moderate Intellectual Disability (ID) exhibit delays in motor development with significant impairment of daily functioning [1,2,3,4]. Indeed, motor skills are functions necessary for Activities of Daily Living (ADL) and adaptive functioning because they allow us to manipulate objects and to move around in the environment [5].

Following the International Classification of Functioning, Disability, and Health (ICF), which postulates a close relationship between body, person, and society as components of personal functioning, both in health and disability conditions, a growing emphasis has been addressed on Physical Activity (PA) for increasing the well-being and social inclusion of the population with ID [6].

### 1.1. Motor Development and Down Syndrome

The link between ID and impaired motor skills has been largely investigated in children and adolescents with Down Syndrome (DS) [7,8,9]. However, very few studies have detected both gross and fine motor competence in adults with DS [10,11,12].

Large-scale research conducted by Smits-Engelsman and Hill (2012) involving 460 children with ID (chronological mean age of 8.9 years and IQ ranging from 50 to 145) showed that children with lower IQ were more likely to show associated lower motor performances [3]. Moreover, the 19% of motor impairments resulted in being explained by the general ID level, and a decrement of 10 percentile points in motor performances was hypothesized for each IQ standard deviation drop. 

Overall, following the “delay hypothesis”, infants and children with DS achieve motor development milestones and motor patterns later than Typically Developing (TD) peers, and this gap increases with age and level of difficulty of motor tasks [13,14,15,16]. For example, Palisano et al. (2001) developed motor growth of gross motor skills such as rolling to prone, sitting on a carpet without support, crawling on hands and knees, standing alone, walking, running, climbing steps, and jumping forward [16]. However, the acquisition of Fundamental Motor Skills (FMS) such as standing alone and walking is mastered between the ages of 18 months and 3 years; running, walking up/downstairs, and jumping between the ages of 3 and 6 years; sitting without support after 14 months; walking unaided after 24–74 months; crawling after 12–18 months. Furthermore, Winders et al. (2018) developed developmental motor schedules based on comparisons among groups of children with DS to identify the reaching of appropriate, advanced, or delayed ages [17]. These schedules included forty-four gross motor skills such as rolling from prone to supine, hand to foot playing, moving from quadruped to sitting laterally, climbing on a sofa, walking with a push toy or without support, and jumping on the floor, riding a tricycle, walking downstairs alternating. These comparisons highlighted a wide variability in the range of achievement age: the mean age was 9.3 months for the rolling from supine to prone, 15 months as concern sitting, 26.8 months as concern pulling to stand from sitting position, 31.3 months as concern stepping 10 ft with 2-hand support, 35.3 months as concern taking 2 independent steps, 40 months as concern walking 15 ft.

Similarly, as concerns the fine motor development in DS, it demonstrated the same trajectory as TD peers, but at delayed ages. Fine motor skills were more impaired than gross motor skills and were characterized by lower developmental increases [13]. For example, children with DS between the ages of 8 and 10 performed worse in motor tasks such as Fine Motor Precision (FMP), Fine Motor Integration (FMI), Manual Dexterity (MD), and Upper Limb Coordination (ULC) [18]. Frank and Esbensen (2015) performed a cross-sectional retrospective chart review on 274 children with DS between the ages of 4 months and 18 years and outlined 3 age groups related to the mastery of fine motor skills such as handwriting, using scissors, and self-feeding/clothing management skills: (1) the “early achievement” if 10–30% of the children with DS mastered the above-mentioned skills; (2) the “representative achievement” if 75–95% of children with DS mastered the skills; (3) the “outliers” if none of the children with DS mastered the skills [14]. Children with DS achieved fine motor skills at delayed ages and with higher variability than their TD peers. For example, they achieved the milestone of raking grasp between 5 and 12 months, string beads between 43 and 144 months, holding a crayon and scribbling between 9 and 36 months, cutting a curved line between 79 and 216 months, holding a bottle independently between 37 and 96 months.

Nevertheless, motor impairments in the DS population refer to restrictions in domains such as motor planning and control, locomotion and object control skill, manual dexterity, fine motor skills, and writing skills [19,20,21,22]. In both children and adults with DS, impaired movement fluency and axial control, hypotonia, and disorders in coordination and body balance, that negatively affect gross motor skills, can be attributed to the decreased size of the corpus callosum and cerebellum, reduced superior temporal gyrus and reduced volume of the brainstem [7,10,23,24,25,26,27,28,29,30,31]. Moreover, in the DS population, physical features such as small hands, short fingers, laxity of ligaments, and lower thumbs can adversely affect fine motor skills such as manipulating small objects and using pencils [11,18,32].

Notwithstanding ample research on motor development in children and adolescents with DS has been produced, studies on motor proficiency in adults with DS are lacking. Nevertheless, it is plausible that adults with DS show impaired motor skills coherently with the decline in their IQ, increasing the difficulty in autonomous skills [33,34]. Consequently, motor impairments negatively affect ADL and adaptive functioning, contributing to a decrease in autonomy, independence, and participation in social activities [35,36]. Furthermore, in adults with DS, an ineffectively developed motor control system does not allow appropriate gross motor skills responses to postural tasks, which is frequently associated with static and dynamic instability accompanied by negative consequences on adaptive functioning [7,10,31]. In a similar way, although there is little research on the topic, the results agree on the lower performance of fine motor skills in adults with DS, negatively impacting ADL [11,32,37]. However, as stated by Vianello (2006), it is fundamental to compare the performances of subjects with DS with those obtained by people of “equivalent age” because this is the real competence mastered in each competence domain and it provides a cleaner index that increases or decreases proportionally in relation to mental age and contextual adaptation [38].

### 1.2. Benefits of Physical Activity

The impaired motor proficiency is associated with an inactive lifestyle which, in turn, contributes to increased health-related complications in DS people [39]. PA is recognized as a key factor in maintaining and improving health during one’s lifespan [40]. Several PA benefits were found in motor proficiency, physical fitness, cognitive function, social domain, and psychological well-being in subjects with DS of all ages [41,42]. The other benefits concern the increase in independence and social inclusion, the improvement of self-esteem, self-competence, self-determination, and self-efficacy [43,44,45,46]. 

Recently, increasing research has demonstrated the relationship between regular PA and brain development, particularly in the prefrontal cortical area, because sport and PA influence the production of neurotrophins, synaptogenesis, and angiogenesis which, consequently, can increase cognitive performances in terms of information processing speed, working memory, and behavioral control strategies [47]. Moreover, as concerns the motivational and emotional domain, motor exercise programs would increase pride and enjoyment, social support, self-confidence, self-esteem, and self-determination [13].

However, most people with DS do not meet the World Health Organization (WHO) recommendations for the practice of PA, which consist of at least 60 min/day of moderate-to-vigorous intensity PA for children and adolescents aged 5–17 years (mostly aerobic), and at least 150 min/week of moderate-intensity aerobic PA, or at least 75 min/week of vigorous-intensity aerobic PA for adults aged 18–64 years [48]. Consequently, a sedentary lifestyle increases the likelihood of gaining weight and the risk of developing obesity, underactive thyroid gland, lower basal metabolic rate, type II diabetes, and heart diseases [49,50,51].

### 1.3. Aim of the Study

In light of these theoretical premises, this study aims to analyze the relationship between motor coordination and global development in subjects with DS, as well as to investigate motor coordination and global development as a function of PA. As concern this second aim, the following specific hypotheses were tested: (H1) motor coordination such as manual dexterity, aiming and catching, and static and dynamic balance were better performed in physically active subjects with DS than in physically inactive subjects with DS; (H2) global development such as physical, adaptive behavior, social-emotional, cognitive, and communication area were better mastered in physically active subjects with DS than physically inactive subjects with DS; (H3) we expect a correlation between motor coordination and global development in subjects with DS and, moreover, greater motor competence in physically active subjects than in physically inactive ones.

The study’s novelty relates to the inclusion of the adult population with DS for the evaluation of these characteristics.

## 2. Materials and Methods

### 2.1. Study Procedure

#### 2.1.1. Participants Recruitment

Participants with DS were recruited through a national public hospital in Palermo city (Sicily, Italy) and two not-for-profit associations that provide support and community resources to people with DS and their families. Medical doctors and educational practitioners from the abovementioned institutions contacted and met with the parents of the subjects with DS to announce the research project and explain the goals and procedures of the study. Subsequently, study researchers asked parents to allow their relatives with DS to take part in the study on a scheduled day. A number of 25 subjects showed up on the day scheduled for the screening and all of them were eligible for the study.

#### 2.1.2. Participants Characteristics

Twenty-five subjects with DS (10 females and 15 males; chronological mean age of 27.24 years; age range: from 17.11 to 46.40 years) and development mean age of cognitive area of 4.93 participated in the study. 

Of these, 15 subjects were attributed to the physically active group (PA-G), given by ≥3 consecutive years of PA practice and according to the WHO recommendations for the practice of PA, while 10 subjects were attributed to the physically inactive group (PI-G), given by ≥3 consecutive years of an inactive lifestyle [48].

The inclusion criteria were as follows: the presence of trisomy 21; the moderate level of ID previously certified by the national public health; the absence of any perceptive disease; the absence of organic impairment; the absence of impairments in independence in stance and ambulation. Participants with DS were excluded from the study if they had mosaicism or trisomy with translocation and a severe level of ID. All participants belonged to medium socioeconomic backgrounds.

#### 2.1.3. Participants Allocation

Following this recruitment phase, all participants were administered a questionnaire on personal data, health status, and PA practice. In particular, in the questionnaire, administered by the same researcher, the participants had to indicate the number of days per week and the number of minutes per day of any type of PA practiced. Subsequently, the same researcher expert in PA and sport, based on the answers obtained, was able to identify whether each participant had reached the minimum levels of PA recommended by the WHO (at least 60 min/day of moderate-to-vigorous intensity PA for participants aged 5–17 years, and at least 150 min/week of moderate-intensity aerobic PA, or at least 75 min/week of vigorous-intensity aerobic PA for participants aged >18 years) in order to allocate the participant in the PA-G [48]. The information on the practice of PA of participants concerned both organized and non-organized PA as long as practiced regularly during the week. Among these activities, sport (both individual and team), exercise (training with an instructor/trainer), and physical activity (leisure physical activity) were included, since regardless of the type of activity, the classification criterion for categorizing the two groups was the achievement of the PA levels recommended by the WHO [48].

#### 2.1.4. Motor Coordination Test Selection

Subsequently, the same researcher with expertise in this research field administered a standardized evaluation to measure participants’ motor coordination, while the parents were given a questionnaire to assess the developmental delays of their relatives with DS.

Regarding the measurement of motor skills, although there is a wide availability of assessments in children, there is a lack of tests in adults. Furthermore, to the best of our knowledge, there are no standardized tests to measure motor coordination in ID. A suggestion is to compare the performances of adults with DS and people with “equivalent age” or “test age” because this would be a cleaner index of the real competencies mastered by each person, increasing or decreasing proportionally in relation to mental age and contextual adaptation [38]. 

Based on these premises, typical population norm-referenced tests are used for the atypical population, without making any attempt to adapt to the condition of ID. For these reasons, we selected a standardized motor coordination evaluation for children because the participants recruited for the study had a development age of cognitive area of 4.93 and on physical area of 4.24. In fact, as reported in the literature, people with moderate ID function at a mental age of about 6–8 years as adults, and they are characterized by difficulties and limitations in basic daily living and in several skills [52]. Moreover, considering the atypical population of the study, we used the raw scores obtained by the participants and not the standardized ones for the typical population.

The study respects the statements of the Declaration of Helsinki for the participation of persons in research, and the Institutional Review Committee of the University of Palermo approved the study (Ref Num 56/2021). Parents provided written informed consent for participation in the study.

#### 2.1.5. Motor Coordination Evaluation

Motor coordination was measured through the Movement Assessment Battery for Children (M-ABC) 2nd edition [53]. This standardized test battery allows to identify and guide treatment of motor impairment in children and adolescents of age ranging from 3 to 16 years. It is composed of 8 fine and gross motor skills subtests clustered into 3 areas: (1) manual dexterity; (2) aiming and catching; (3) static and dynamic balance. Tasks corresponding to the age band 11.0–16.11 were employed. Manual dexterity included performing tasks such as turning pegs, building triangles with nuts and bolts, and drawing trails, aiming and catching included tasks such as catching with one hand and throwing at a wall target. Static and dynamic balance took into account tasks such as two-board balance, walking toe-to-heel backwards and zigzag hopping.

The score was binary 0 or 1. The raw scores, such as completion time in secs, number of errors/correct, catches/correct, throws/steps/jumps/hops (maximum total score: 48), were computed by adding the scores items pertaining to each category with high scores indicating poor performance. A global motor coordination raw score was obtained by adding scores for each category. We used the raw scores without normalizing them by age because, according to the literature, we considered the mental age and not the chronological age for comparisons. It is appropriate to compare the performances of people with DS and people of “equivalent age” [38].

After giving the assignment, which was based on the Examiner’s Manual guidelines, the experimenter showed the exercise to ensure correct understanding by the participants. Verbal reinforcement was given after each task was performed.

All participants attended tasks without any resistance and showed full cooperation.

The assessment was individual, took place in a quiet room, and lasted about 25–30 min.

As concern the psychometric properties, the internal consistency was Cronbach’s α = 0.83 for the total scale, α = 0.59 for the manual dexterity subscale, α = 0.79 for the aiming for and catching subscale, and α = 0.91 for the balance domain subscale.

#### 2.1.6. Global Development Evaluation

Global development was measured through the Italian version [54] of the Developmental Profile 3 (DP-3) [55]. It is a parent checklist to screen developmental strengths and weaknesses from birth to 12 years, which includes the following 5 areas: (1) physical area (35 items, e.g., “Does he usually go up and down stairs with only one foot on each step?”); (2) area of adaptive behavior (37 items, e.g., “Does he usually wash his face and hands and dry them unaided?”); (3) social-emotional area (36 items, e.g., “Does he show to understand the moods of others by saying things like “he’s very angry”, “he’s angry”, “he’s afraid”, or “throwing a tantrum?”); (4) cognitive area (38 items, e.g., “On request, is he able to remove 13 objects from a group of 20?”); (5) communication area (34 items, e.g., “Does he talk to peers for at least 1 hour most days?”). 

Each scale has start points corresponding to the chronological age. Parents were asked to indicate if their relatives with DS mastered (score = 1) or not (score = 0) skills related to the abovementioned areas. The raw scores were computed by adding the scores items pertaining to each area. A global development raw score was obtained by adding scores for each area.

The assessment took place in a quiet room.

The Italian translation used proves to have a reliability (α = 0.68) equal to that of the original version in English (α = 0.68).

### 2.2. Statistical Analysis

A priori sample size power analysis was calculated using G*Power software 3.1.9.2 (Heinrich Heine University, Düsseldorf, Germany).

The distribution of normality was checked through the Shapiro-Wilk test.

Since data were not normally distributed, a non-parametric analysis was performed. Firstly, a descriptive statistic of all the variables was carried out. 

A Spearman’s Rank-Order correlation coefficient was calculated to measure any relationship between motor coordination skills and global development areas in the entire sample.

A Mann–Whitney *U* test was computed in order to assess any differences in motor coordination and global development between the two groups (i.e., physically active and physically inactive participants).

Scatterplots and boxplots were used to represent the significant correlations and differences found, respectively.

The software Statistica 12 (StatSoft^®^, TIBCO^®^ Software Inc, Palo Alto, CA, USA) was used to perform all the statistical analyses with the significance set at *p* < 0.05.

Moreover, a post-hoc sample size power analysis was computed using G*Power software 3.1.9.2 (Heinrich Heine University, Düsseldorf, Germany).

## 3. Results

The a priori sample size power analysis (1-tail, d = 0.5, α = 0.05) revealed that the total sample size required to obtain a power of 0.80 should be 102 participants (i.e., 51 per group).

Descriptive statistics of the entire sample for both the M-ABC and the DP-3 variables are reported in Table 1. 

Table 2 shows means and SD of the raw scores of all the variables considered for the PA-G and the PI-G.

The Spearman’s correlation coefficient among the M-ABC and the DP-3 variables showed positive correlations between: the global motor coordination and the global development (ρ = 0.56, *p* = 0.002) (Figure 1), the global motor coordination and the adaptive behavior development area (ρ = 0.43, *p* = 0.021) (Figure 2), the aiming and catching skills and the global development (ρ = 0.50, *p* = 0.006) (Figure 3), the aiming and catching skills and the adaptive behavior development area (ρ = 0.41, *p* = 0.023) (Figure 4). In Table 3, the results of correlation analyses are displayed.

As presented in Table 2 and Table 4, PA-G showed higher values for all subtests of both the M-ABC and the DP-3 compared to PI-G, although, through the Mann–Whitney *U* test, significant differences were found only for the global raw score of the M-ABC (Z = 2.504, *p* = 0.012) (Figure 5), and for the aiming and catching skills (Z = 2.248, *p* = 0.025) (Figure 6), as well as for the physical development area of the DP-3 (Z = 1.980, *p* = 0.048) (Figure 7).

The post-hoc sample size power analysis (1-tail, d = 0.5, α = 0.05) revealed a power of 0.33.

## 4. Discussion

The first aim of this study was to investigate the relationship between coordinative proficiency and global development in subjects with DS including adults.

On the whole, significant positive correlations were found between global motor coordination and global development, and moreover between global motor coordination and the adaptive behavior development area. Similarly, we found significant positive correlations between the aiming and catching skills and global development, and moreover between the aiming and catching skills and the adaptive behavior development area. These results are well documented in the literature showing how motor skills and postural patterns are related to adaptive functioning, independence, and participation in social activities [35,36]. Indeed, fine motor skills are a key factor in carrying out ADL and allow us to successfully engage in daily self-care activities [56,57,58]. However, we found a moderate magnitude of the correlation between the adaptive behavior development area and the global motor coordination, and the aiming and catching skills, probably because in adults with ID, the IQ decline corresponds to the increase in difficulty in autonomous skills and the weakening of daily functioning [1,2,3,4,33,34].

The second aim of the current study was to investigate motor coordination skills and global development as a function of PA by the hypothesis that physically active subjects with DS would show higher levels of performance than physically inactive subjects with DS.

Regarding motor coordination, subjects with DS who regularly practice PA performed higher scores than physically inactive subjects with DS in all the tasks. However, significant differences were found only in the global motor coordination and in the aiming and catching skills. These results are coherent with physically active DS participants’ higher motor competencies that were reported by their parents compared to physically inactive participants. Thus, better motor competencies were demonstrated by adults with DS practicing PA with both the measures. These measures involve two approaches, such as long-term information on fine and gross motor proficiency in an ecological context, derived from a performance test as the M-ABC, and a more exhaustive assessment, derived from a parent’s checklist as the DP-3. 

Similarly, participants with DS belonging to the PA-G reported higher levels of global development and higher scores for each area. Indeed, higher motor performances in physically active subjects are documented in the literature showing that PA practice is a key factor to improve health, cognitive function, and performances of daily living activities in adults with DS [42,59]. Moreover, the processing of information and actions-based representation in PA is able to stimulate the emergence of higher-order representation and knowledge acquisition [60]. Scifo et al. (2019) in their review detailed the beneficial effects of PA in the physical and mental domains [40]. The first ones concern motor proficiency, physical fitness, bone metabolism, cardiovascular and respiratory muscle functions, obesity control, and prevention of coronary heart diseases. Furthermore, it is well known that PA improves postural control reducing the risk of falls [61,62]. The health benefits of PA are widely recognized and, a growing body of evidence reported that the practice of PA during the midlife period is fundamental for healthy aging [63,64]. The second one concerns the increase in independence and social inclusion, the improvement of self-esteem, self-competence, self-determination, and self-efficacy. 

Research has shown that the regular practice of PA affects the production of neurotrophins, synaptogenesis, and angiogenesis which, as a consequence, is likely to increase cognitive abilities such as information processing speed, working memory, and behavioral control strategies [47]. In particular, better performance in the aiming and catching subtest of the PA-G is an interesting example of PA benefits. Aiming and catching skills are complex goal-directed tasks with a high motor and cognitive load because they involve motor skills such as speed, hand/arm/leg coordination, and explosive strength, as well as cognitive abilities such as visual search, visual discrimination, and focal attention, shifting of the attentive focus, working memory, and decision making [65]. For these reasons, the lower motor skills performance of the PI-G could be due to the lack of PA practice in these participants. Indeed, as is widely recognized, although the WHO recommends at least 150 min/week of moderate-intensity aerobic PA, or at least 75 min/week of vigorous-intensity aerobic PA for adults aged 18–64 years, most adults with DS do not meet these recommendations [48]. Hence, in subjects with DS, the health-related PA benefits are limited by the low rate of PA practice. However, in agreement with several studies in the literature [48,49], we did not show significant differences between PA-G and PI-G in specific tasks of motor coordination (manual dexterity and static and dynamic balance) and global development (adaptive behavior area, social-emotional area, cognitive area, and communication area) because in subjects with DS the health-related PA benefits in the above-mentioned areas could be due to the specificity and quality of the PA performed. The analysis of the determinants of PA in subjects with DS is still ongoing. Following the approach of the ICF [49], the effectiveness of PA could be affected by a person’s functional profile, health status, involvement in life activities, and contextual factor (within the person or in the environment).

For these reasons, it is realistic to assume that health complications may present barriers to the practice of PA in subjects with DS and that, when clinical features are efficiently overcome, PA can be facilitated.

Moreover, a further explanation could be linked to the inclusion in the PA-G of those children with DS who had achieved the minimum levels of PA recommended by the WHO. This allowed us to include subjects participating in a wide range of PA, exercise, or sport (from leisure PA to individual/team sport). The subjects of the PA-G had, therefore, not carried out specific training protocols for the improvement of manual dexterity, body balance, adaptive behavior area, social-emotional area, cognitive area, and communication area. Nevertheless, we detected that the practice of the minimum levels of PA recommended by the WHO overall induced higher scores in motor coordination and global development in physically active subjects with DS than in physically inactive ones.

### Strength and Limitations of the Study

To the best of our knowledge, only sparse studies have previously investigated the motor competence in adults with DS in association with global development in cognitive and adaptive areas and this represents the major strength of the study. However, the most important limitation of the current study lies in methodological concerns as the relatively small sample size and the measures carried out. In effect, although the sample represents one of the main limitations of the study, it is known that the recruitment of large numbers of participants in clinical populations (and in particular those who practice PA) is one of the greatest difficulties that is experienced. Furthermore, the pandemic situation did not allow us to extend the sample size. Regarding methodological concerns, although there is a lack of motor coordination tests for adults with DS it should be noted that: (1) we used a standardized motor coordination assessment for children because the participants had a moderate level of ID (and people with moderate ID function at a mental age of about 6–8 years as adults); (2) we used the raw scores obtained by the participants and not the standardized ones for the typical population. Finally, another limitation of the study that should be mentioned is the moderate magnitude of the correlations we found. This finding could be primarily explained by the small sample size we recruited.

## 5. Conclusions

To sum up, the findings of this study reinforce the need to implement and encourage the practice of PA in order to promote well-being and social inclusion not only, as it is already known, in children and adolescents but also in adults with DS. Moreover, researchers and practitioners are encouraged to implement PA to enhance motor and cognitive skills in DS adults, as well as their sense of social inclusion and ADL, and belonging, self-determination, and self-esteem [66,67,68,69,70]. All of these factors are expected to have a positive effect on adaptive functioning and ADL which, in turn, improve autonomy and independence [13]. The context of the present research and the related results are of practical use for medical doctors, psychologists, special educators, and kinesiologists. Moreover, these findings add new knowledge for health psychoprophylaxis for adults with DS.

## Figures and Tables

**Figure 1 jcm-11-05031-f001:**
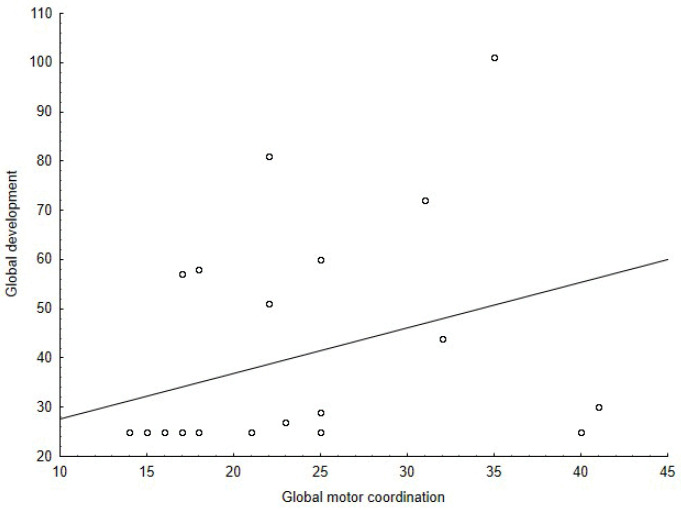
Correlation between global development of the DP-3 and global motor coordination of the M-ABC.

**Figure 2 jcm-11-05031-f002:**
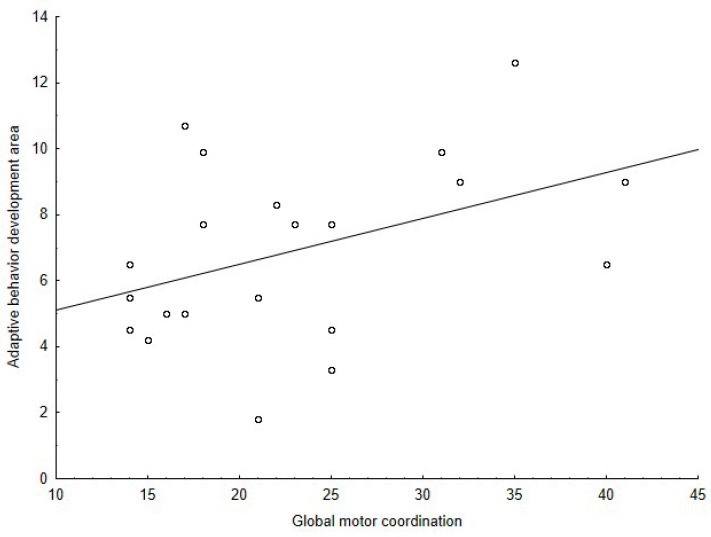
Correlation between adaptive behavior development area of the DP-3 and global motor coordination of the M-ABC.

**Figure 3 jcm-11-05031-f003:**
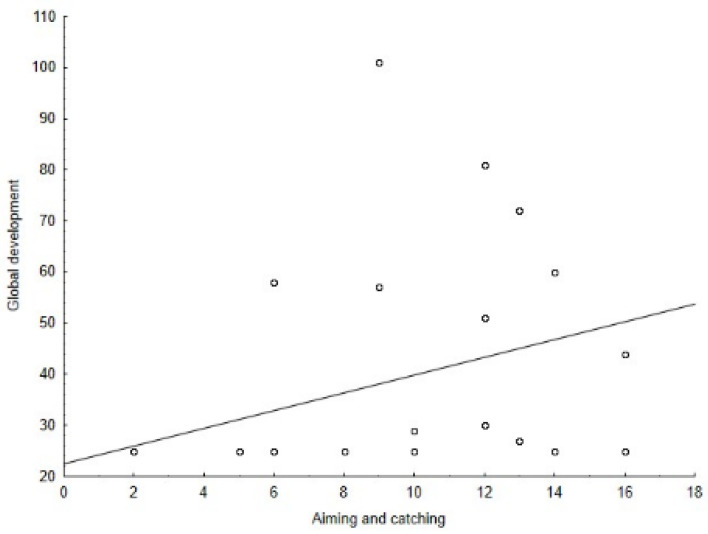
Correlation between global development of the DP-3 and aiming and catching skills of the M-ABC.

**Figure 4 jcm-11-05031-f004:**
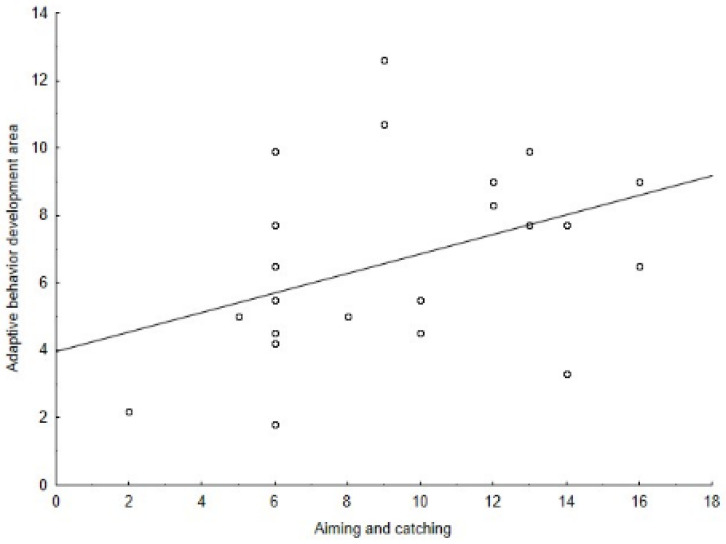
Correlation between adaptive behavior development area of the DP-3 and aiming and catching skills of the M-ABC.

**Figure 5 jcm-11-05031-f005:**
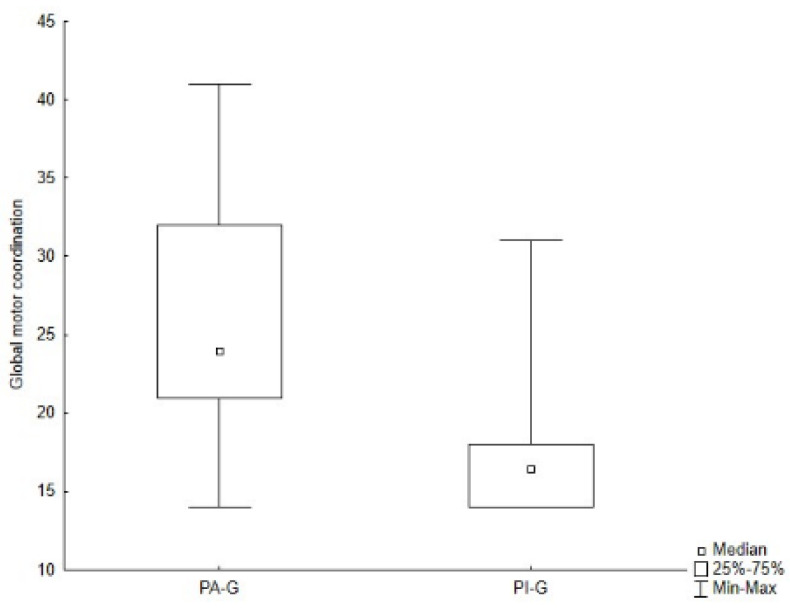
Difference in global motor coordination of the M-ABC between groups. Legend. PA-G, Physically Active Group; PI-G, Physically Inactive Group.

**Figure 6 jcm-11-05031-f006:**
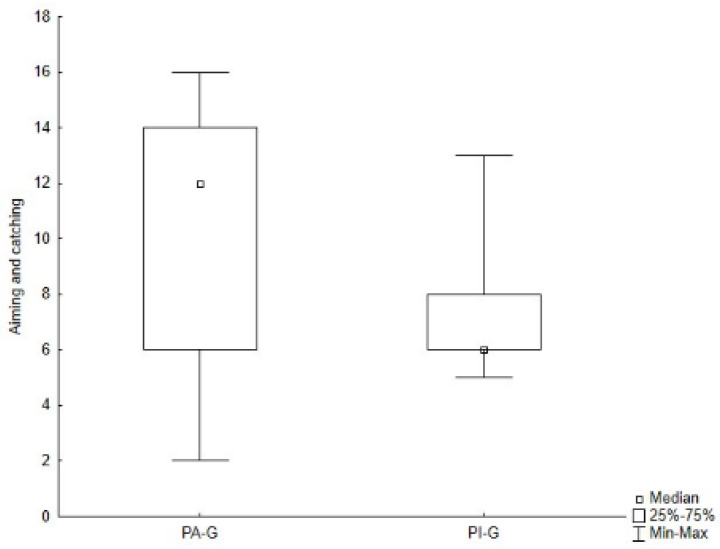
Difference in aiming and catching skills of the M-ABC between groups. Legend. PA-G, Physically Active Group; PI-G, Physically Inactive Group.

**Figure 7 jcm-11-05031-f007:**
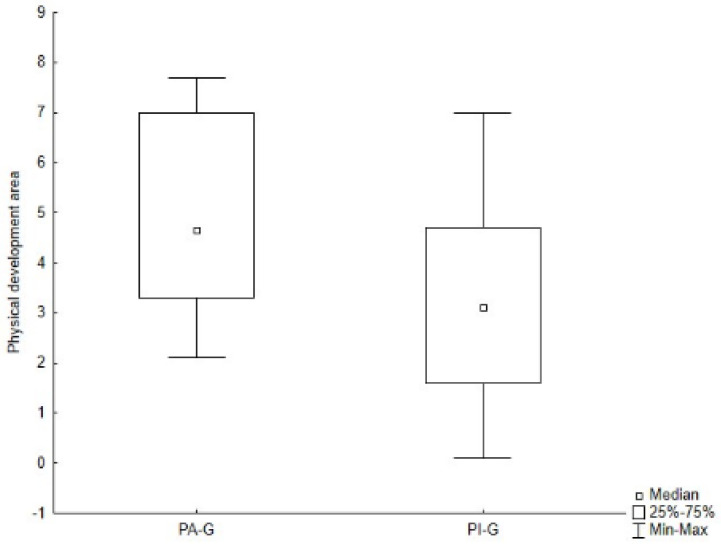
Difference in physical development area of the DP-3 between groups. Legend. PA-G, Physically Active Group; PI-G, Physically Inactive Group.

**Table 1 jcm-11-05031-t001:** Descriptive statistics of the entire sample.

Variable	Min	25th p	Median	Mean	SD	75th p	Max
**M-ABC**							
Manual dexterity	3.0	3.0	3.0	3.72	2.39	3.0	14.0
Aiming and catching	2.0	6.0	9.0	9.16	3.85	12.0	16.0
Static and dynamic balance	5.0	5.50	8.0	9.17	5.06	10.0	26.0
Global motor coordination	14.0	15.50	21.0	22.25	8.21	25.0	41.0
**DP-3**							
Physical area	0.11	2.36	4.11	4.24	2.21	7.0	7.70
Adaptive behavior area	1.80	4.50	6.50	6.66	2.76	8.65	12.60
Social-emotional area	1.80	3.05	4.10	5.14	2.84	7.16	10.80
Cognitive area	1.40	4.0	4.80	4.93	1.68	5.86	8.20
Communication area	1.10	2.80	4.10	4.49	2.47	5.31	10.20
Global development	25.0	25.0	25.0	38.40	21.46	51.0	101.0

Legend. M-ABC, Movement Assessment Battery for Children; DP-3, Developmental Profile 3; Min, Minimum; 25th p, 25th Percentile; SD, Standard Deviation; 75th p, 75th Percentile; Max, Maximum.

**Table 2 jcm-11-05031-t002:** Descriptive statistics of the PA-G and the PI-G.

Variable	PA-G Means ± SD	PI-G Means ± SD
**M-ABC**		
Manual dexterity	3.73 ± 2.84	3.70 ± 1.64
Aiming and catching	10.53 ± 4.09	7.10 ± 2.38
Static and dynamic balance	10.57 ± 6.07	7.20 ± 2.20
Global motor coordination	25.50 ± 8.57	17.70 ± 5.17
**DP-3**		
Physical area	5.00 ± 2.05	3.18 ± 2.07
Adaptive behavior area	7.21 ± 2.73	5.88 ± 2.74
Social-emotional area	5.93 ± 3.21	4.02 ± 1.86
Cognitive area	5.02 ± 1.65	4.81 ± 1.81
Communication area	4.77 ± 2.50	4.10 ± 2.50
Global development	42.07 ± 23.81	32.90 ± 17.03

Legend. M-ABC, Movement Assessment Battery for Children; DP-3, Developmental Profile 3; PA-G, Physically Active Group; PI-G, Physically Inactive Group; SD, Standard Deviation.

**Table 3 jcm-11-05031-t003:** Correlation analysis among the M-ABC and the DP-3 variables through Spearman’s correlation coefficient.

		M-ABC			
		Manual Dexterity	Aiming and Catching	Static and Dynamic Balance	Global Motor Coordination
**DP-3**	Physical area	ρ = 0.23, *p* = 0.138	ρ = 0.32, *p* = 0.06	ρ = 0.10, *p* = 0.320	ρ = 0.31, *p* = 0.078
	Adaptive behavior area	ρ = 0.20, *p* = 0.176	**ρ = 0.41, *p* = 0.023**	ρ = 0.24, *p* = 0.135	**ρ = 0.43, *p* = 0.021**
	Social-emotional area	ρ = 0.15, *p* = 0.243	ρ = 0.34, *p* = 0.05	ρ = 0.06, *p* = 0.391	ρ = 0.24, *p* = 0.131
	Cognitive area	ρ = 0.17, *p* = 0.213	ρ = 0.10, *p* = 0.327	ρ = 0.04, *p* = 0.422	ρ = 0.09, *p* = 0.340
	Communication area	ρ = 0.23, *p* = 0.136	ρ = 0.34, *p* = 0.05	ρ = −0.02, *p* = 0.467	ρ = 0.25, *p* = 0.122
	Global development	ρ = 0.34, *p* = 0.05	**ρ = 0.50, *p* = 0.006**	ρ = 0.29, *p* = 0.081	**ρ = 0.56, *p* = 0.002**

Legend. M-ABC, Movement Assessment Battery for Children; DP-3, Developmental Profile 3.

**Table 4 jcm-11-05031-t004:** Comparisons between the PA-G and the PI-G through Mann–Whitney *U* Test.

Variable	Rank Sum PA-G	Rank Sum PI-G	Z	*p*
**M-ABC**				
Manual dexterity	186.0	139.0	−0.835	0.404
Aiming and catching	235.0	90.0	2.248	**0.025**
Static and dynamic balance	197.50	102.50	1.304	0.192
Global motor coordination	218.0	82.0	2.504	**0.012**
**DP-3**				
Physical area	209.0	91.0	1.980	**0.048**
Adaptive behavior area	196.0	104.0	1.204	0.223
Social-emotional area	199.50	100.50	1.409	0.159
Cognitive area	175.50	124.50	0.000	0.775
Communication area	187.0	113.0	0.676	0.527
Global development	222.0	103.0	1.618	0.106

Legend. M-ABC, Movement Assessment Battery for Children; DP-3, Developmental Profile 3; PA-G, Physically Active Group; PI-G, Physically Inactive Group.

## Data Availability

The data presented in this study are available on request from the corresponding author.

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
