# Peer review of "Motor Coordination and Global Development in Subjects with Down Syndrome: The Influence of Physical Activity"

_jcm, 2022, doi:10.3390/jcm11175031_

Round 1
Reviewer 1 Report
I have reviewed the article entitled “Motor coordination and global development in subjects with 2 Down Syndrome: The influence of physical activity". There is a large gap in knowledge regarding the value of physical activity in Down syndrome, so the need to better understand the impact of physical activity across a spectrum of Down syndrome individuals social interactions, well-being and function is extremely relevant. This manuscript provides the first experimental measurements to try and identify if physical activity benefits exist in Down syndrome.
Questions remain:
1. Concerns with power are valid. Introduce these in the methods section, explaining it at the end also helpful, but what numbers of Down syndrome individuals would be needed to see significant differences? This calculation is now possible
2. Please explain more about the individual tests performed and how Down syndrome individuals managed them
Other concerns:
1. Please have some strong editorial and grammatical review.
2. Box-whiskers plots were difficult to read. What is the median 25%-75% box (e.g. figure 6,7)? Almost illegible
Reviewer 2 Report
Thank you for the opportunity to review the manuscript titled:
Motor coordination and global development in subjects with Down Syndrome: The influence of physical activity
It is an interesting article with a good design that aims to present the analysis of the relationship between motor coordination and global development in subjects adults with Down Syndrome.
It is a very important contemporary topic because motor coordination and global development in adults whit Down Syndrome its many dimensions, is an area that needs nuanced research (medically, psychological and psychoprophylaxis) – this paper sets out to address this in part by looking at the influence of physical activity.
The article provides significant scientific support for the transmission of the results of these studies to the practice of work of health care professionals, psychologist or educators.
It is also a great complement to the literature on the subject and can be considered for publication in this journal and presenting to the readers in Special Issue.
However, there are some small details that should be improved prior to publication:
Aim of study
• A separate section should be Introduced in the text of the article, where the aim of the study and research problems will be presented. It is worth organizing theoretical part of article exposing the main methodological part of the work in a separate part of the article.
•I propose for the part titled Purpose (line 146-160) to be renamed Aim of study. Additionally, specific hypotheses (line 150-155) should be highlighted as H1, H2. The entry in lines 158-160 is the hypothesis H3 - please add it as H3 - especially since the authors report its results in the Results part (lines: 293-298).
Materials and Methods
The instruments, sample and statistical analysis conducted are appropriate.
• Please supplement the provision with the psychometric properties of the research instruments used.
• Please complete the procedure for providing feedback on the test results to the respondents.
Results
The results are properly analyzed but the presentation of the results has some problems.
I encourage the authors of the article to consider supplementing the Results section with the following items:
• The p-values and key statistics for all tests must be reported. Ideally, they would all be reported in the manuscript. It is inappropriate to only report significant effects. Please complete in table 3 entry for p, which was reported as n.s.
• Please add table / tables with correlations presented in graphs 1, 2, 3 and 4. Additionally, it is worth using specific p-values (not p <.05) for entry this correlations
Discussion:
Impressive and discussed with all the issues.
However, it is recommended to deep dive on issues in discussion - mainly the practical context of research, especially that may be useful in the work of doctors, psychologists, special educators or broadly understood for health psychoprophylaxis for adults with Down Syndrome
Limitations
•Proposes to add a separate Limitations subsection. Currently, the description of the limitations of the research is part of the discussion (line 426-441)
Conclusions
•Proposes to add a separate subsection Conclusions. Currently, the description of the Conclusions of the research is part of the discussion (line 418-425)
References
Please update the reference list.
I hope that the suggested changes help to improve the quality of the article and that they are well received.
Kind regards
